# Multimodular Bio-Inspired Organized Structures Guiding Long-Distance Axonal Regeneration

**DOI:** 10.3390/biomedicines10092228

**Published:** 2022-09-08

**Authors:** Laura Rodríguez Doblado, Cristina Martínez-Ramos, Manuel Monleón Pradas

**Affiliations:** 1Center for Biomaterials and Tissue Engineering, Universitat Politècnica de València, 46022 Valencia, Spain; 2Unitat Predepartamental de Medicina, Universitat Jaume I, 12071 Castellón de la Plana, Spain; 3Biomedical Research Networking Center in Bioengineering Biomaterials and Nanomedicine, CIBER-BBN, 28029 Madrid, Spain

**Keywords:** long-distance axonal regeneration, hyaluronic acid conduit, poly-lactic fibers, dorsal root ganglion culture, Schwann cell culture, neural cord

## Abstract

Axonal bundles or axonal tracts have an aligned and unidirectional architecture present in many neural structures with different lengths. When peripheral nerve injury (PNI), spinal cord injury (SCI), traumatic brain injury (TBI), or neurodegenerative disease occur, the intricate architecture undergoes alterations leading to growth inhibition and loss of guidance through large distance. In order to overcome the limitations of long-distance axonal regeneration, here we combine a poly-L-lactide acid (PLA) fiber bundle in the common lumen of a sequence of hyaluronic acid (HA) conduits or modules and pre-cultured Schwann cells (SC) as cells supportive of axon extension. This multimodular preseeded conduit is then used to induce axon growth from a dorsal root ganglion (DRG) explant placed at one of its ends and left for 21 days to follow axon outgrowth. The multimodular conduit proved effective in promoting directed axon growth, and the results may thus be of interest for the regeneration of long tissue defects in the nervous system. Furthermore, the hybrid structure grown within the HA modules consisting in the PLA fibers and the SC can be extracted from the conduit and cultured independently. This “neural cord” proved to be viable outside its scaffold and opens the door to the generation of ex vivo living nerve in vitro for transplantation.

## 1. Introduction

Axonal bundles or axonal tracts possess an aligned and unidirectional architecture that is present in many neural tissues where communication across distances of different lengths is needed, both in the central and in the peripheral nervous system. Loss of communication between the central nervous system (CNS) and the peripheral nervous system (PNS) is the cause of countless disorders that compromise the quality of life of affected people. This loss of communication can be due to peripheral nerve injury (PNI), spinal cord injury (SCI), traumatic brain injury (TBI), or neurodegenerative diseases, such as Parkinson’s disease. When one of them occurs, the intricate architecture of the nervous tissue undergoes alterations leading to growth inhibition and loss of guidance [1,2].

Injuries in peripheral nerves can result in a gap between two nerve stumps. When nerve endings are unable to rejoint without tension, a bridging section of the nerve is used, and two end-to-end sutures are performed [3]. In this case, a nerve graft is used, usually from the patient (autograft) or tissue from a cadaveric human donor (allograft). However, the use of autografts has limitations [4]. Allografts overcome several of the drawbacks of autografts but require immunosuppression or decellularization to prevent immune rejection [5]. These problems drive the search for a tissue engineering alternative to this treatment.

The Food and Drug Administration (FDA) has approved several conduits based on natural and synthetic degradable biomaterials to repair nerve defects arising from PNI [6], all limited to use in relatively short distance peripheral nerve reconnection but with advantages in comparison with autografts [4]. However, in most cases, these nerve conduits are not as good as the autograft. In general, failure of nerve regeneration across long gaps seems to be the result of a lack of the formation of an initial fibrin cable, which is necessary for Schwann cell (SC) migration into the constructs and the formation of the bands of Büngner, which are aligned columns of SC and laminin [7]. There is a limitation in nerve regeneration if the gap is >10 mm in rats [8]. Synthetic conduits have only been used successfully clinically to bridge the injured nerve stumps when the gaps are <10–12 mm [9], but recently researchers have achieved promising results in the rat sciatic nerves [10]. Most research focuses on filling the lumen of nerve guides with scaffolds, fibers, cells, and/or drug delivering therapies in order to improve the regeneration capacities in long gap PNI [11], such as porous cylindrical collagen structures whose ellipsoidal cavities have a preferential direction [12], or aligned polymer fiber-based constructs [13]. However, the most common is the use nerve guidance conduits with one or more parallel channels [14,15,16] that serve as a preferential path or guide to the axons, or combined systems of the above [17].

In the case of CNS regeneration, there is no clinical treatment with a proven recovery. In SCI, TBI, and degenerative disease, initial damage results in fast acute primary injury events. Current treatments, instead of trying to recover damaged tissue, try to prevent secondary injury [18]. Tissue engineering strategies make use of different biomaterials with channels or guides [19,20,21]. With the same guiding cues idea for the treatment of brain lesions, researchers propose tubular micro-columns containing hydrogel with the capacity of astroglia cells to create a neural guide pathway [22] or a more complex micro-column system containing mature primary cortical neurons and long axonal tracts proposed as a possible treatment for Parkinson’s disease [23].

SC are key players in the neural repair process in the PNS and constitute the necessary auxiliary cell support for axon outgrowth. SC can secrete neurotrophic factors, which is important for nerve regeneration, as well as cell adhesion molecules [24]. The SC supporting regeneration effect in vitro [25] and in vivo [26] has been consistently reported, and different approaches have been employed in combination with materials in order to induce an alignment to orient in vitro neuronal growth [27,28]. Moreover, nerve guides pre-filled with SC have been proven to support regeneration in rat PNS [29] and CNS [30].

In our previous studies [31,32,33,34,35], we found that SC seeded into hyaluronic acid (HA) conduits with poly-L-lactide acid (PLA) microfibers in their lumina were able to proliferate and self-organize into a continuous cylindrical cell sheath in the lumen of the conduits. Additionally, SC grew on the PLA fibers, spanning the whole distance of a biohybrid construct 6 mm in length. We demonstrated that the biohybrid construct proved effective in promoting directed axon growth along the entire 6 mm length when a DRG explant was placed at one of its ends. HA is a natural glycosaminoglycan, an essential component of the extracellular matrix (ECM) of many tissues [36], and it is biocompatible and biodegradable. HA has already shown therapeutic benefits on neuronal regeneration processes [37,38] and exhibits mechanical properties similar to soft nervous tissues [39]. PLA is a synthetic polyester with mechanical properties, biodegradability, and biocompatibility [40]. PLA fibers raise interesting questions as a part of the tubular conduit concept when allocated in the lumen [41], providing support for cell adhesion, migration, and elongation in a guided way.

Typical biomaterial conduits have limited lengths, and the span or gap to be regenerated may be greater than those lengths. Here, we explore the idea of building a long scaffold for axon tract regeneration out of several shorter modules. This multimodular concept consist of a PLA fiber bundle placed within the common lumen of several individual HA conduits positioned consecutively, designing nerve guidance conduits with desired length by changing the number of HA modules. We compare the multimodular construct with a unimodular equivalent scaffold (Figure 1). We seeded and cultured SC into an 18 mm unimodular or multimodular conduit for 5 days. Then, we put at one end of the construct a dorsal root ganglion (DRG) explant, as a source of projecting neurons, and the axon extension stemming from this source into the SC-seeded conduit lumen was observed (Figure 1 bottom). We compare the multimodular conduit with a unimodular conduit with the same length to study the cells viability and distribution, and the axonal extension of neurons coming from the DRG explant. In a separate study [35], we adapted the present multimodular concept for an in vivo experimental model of sciatic nerve regeneration in rabbits, which we discuss in connection with the present findings below.

Moreover, after these 5 days of SC culture, we extracted from the HA scaffold the hybrid fiber-cells structure that had developed within; this hybrid ‘neural cord’ developed in the interior of the multimodular conduit and was then independently cultured for 5 additional days (Figure 2). With the development of this stable ‘neural cord’, we can obtain a live hybrid pseudonerve in vitro for possible transplantation.

## 2. Materials and Methods

### 2.1. Cell Source

Primary rat Schwann cells (SC; Innoprot, Bizkaia, Spain) expanded with Schwann Cell Medium (Innoprot, Bizkaia, Spain) were employed at 4–5 cell passage for cell cultures in the materials. Sprague–Dawley rats from Charles River were used for DRG explant dissection and were maintained following the National Guide to the Care and Use of Experimental Animals (Real Decreto 1201/2005). After the sacrifice by decapitation, whole DRG explants were dissected from the spinal column of neonatal rats (P3–P4) and transferred into ice-cooled Dulbecco’s modified Eagle medium with a high glucose level (4.5 g/L) (DMEM; Thermo Fisher Scientific, Madrid, Spain) containing 10% fetal bovine serum (FBS; Thermo Fisher Scientific, Madrid, Spain), using a dissecting microscope to remove the remaining nerves and connective tissue.

### 2.2. Preparation of Hyaluronic Acid Unimodular and Multimodular Conduits with Poly-L-Lactic Acid Fibers for In Vitro Experiments

The synthesis of HA sodium salt from Streptococcus equi (HA; 1.5–1.8 MDa; Sigma-Aldrich, Madrid, Spain) conduits was carried out as previously described [31,32,33,34,35]. Briefly, poly-ε-caprolactone (PCL; Mw = 40 kDa; PolySciences, Madrid, Spain) fibers of 400 µm were extruded in Hater Minilab, and a polytetrafluoroethylene thin block with grooves 1.5 mm wide with a single PCL fiber of 400 µm diameter was used as a mould for the conduits. A total of 5% (*w*/*v*) of HA was dissolved for 24 h in sodium hydroxide 0.2 M (Scharlab, Madrid, Spain). Then, the HA was crosslinked with divinyl sulfone (DVS; Sigma-Aldrich) in a 9:10 DVS:HA monomeric units molar ratio and this solution was mixed and injected in a mould. Later, the solution in the mould was lyophilized for 24 h (Lyoquest-85, Telstar, Bensalem, PA, USA) to generate HA microporous matrices. Finally, the conduits were hydrated in distilled water for 2 h, the PCL fibers were extracted, and the conduits were cut to the desired length. In several conduits, 120 aligned PLA (AITEX Textile Research Institute, Valencia, Spain) fibers of 10 µm diameter were placed inside the channel of the conduits. In order to obtain the unimodular conduit, a PLA fibers bundle was introduced in a long unimodular conduit (12 mm or 18 mm length). To form the multimodular conduit, a PLA fibers bundle was introduced in the lumen of three shorter individual conduits or modules (4 mm or 6 mm length), which positioned themselves one behind the other. Fixing structures were added at both ends of the PLA fibers bundle that protrudes from the conduits to hold the unimodular conduit and individual modules in the multimodular conduit together. In this work, we have opted to use three individual modules for the multimodular conduit and thus have two inter-module zones. We have not chosen to work with more individual modules due to the long culture time that this would entail.

Before the seeding of the cells, the unimodular and multimodular conduits were sanitized for 2 h with 70° ethanol (Scharlab, Madrid, Spain) and 10 min with 50°, 30° ethanol, and distilled water. Then, conduits were conditioned with culture medium overnight.

### 2.3. Cell Culture and Cell Seeding within Conduits

We used 18 mm length unimodular conduits and multimodular conduits of three 6 mm length modules for all the experiments with cell culture. SC 4–5 cell passage were grown in flasks, until confluence at 37 °C, 5% CO_2_, in SC medium. SC were seeded at a density of 600,000 cells in a total length of 18 mm (unimodular or multimodular conduit) and 9 µL of SC medium. SC were seeded with a 1–10 micropipette—inserting the tip at one end on each individual module that formed a multimodular conduit (3 µL in each module) or in the two ends in the case of the unimodular conduit (4.5 µL in each end) to ensure the homogeneous distribution of cells in the lumen—and were maintained in the incubator for 30 min before adding the SC medium. SC were cultured for 5 days in SC medium. After that, DRG explants were placed in direct contact with one end of the unimodular or multimodular conduits and were then transferred into 6-well plates, which were maintained in a tissue culture incubator at 37 °C, providing a humidified atmosphere containing 5% CO_2_ with a specific DRG medium (Neurobasal medium (Thermo Fisher Scientific, Madrid, Spain), D-glucose 2 mg/mL (Sigma-Aldrich, Madrid, Spain) L-glutamine 100× (Thermo Fisher Scientific, Madrid, Spain), 1% FBS, 1% penicillin/streptomycin (Thermo Fisher Scientific, Madrid, Spain), 2% B27 supplement (Thermo Fisher Scientific, Madrid, Spain), 0.1% nerve growth factor (NGF; Thermo Fisher Scientific, Madrid, Spain)) refreshed every 2 days until 21 additional days counted from the moment the DRG was seeded.

### 2.4. Development of a ‘Neural Cord’

After 5 days of the SC culture being inside the multimodular conduits, using a stereoscope for visual guidance, we made a longitudinal cut with a scalpel of the different modules to eliminate the HA conduits protection to obtain a complete ‘neural cord’ formed by the PLA fibers and the associated SC. The ‘neural cord’ was transferred with sterile surgical forceps into a 6-well plate and cultured for 5 days more in SC medium.

Moreover, a proof of concept was performed seeding DRG explants to confirm that it is possible to grow and guide the axons through the ‘neural cord’. After 5 days of SC culture, a DRG explant was placed in one end of the multimodular conduit and cultivated for 10 days with DRG medium. Then, we eliminated the HA conduits and cultivated the ‘neural cord’ with the DRG explant for 16 additional days.

### 2.5. Scanning Electron Microscopy

The unimodular and multimodular (three 4 mm length modules) conduits with a total length of 12 mm were dried for scanning electron microscopy (SEM) visualization. The conduits were cut longitudinally to expose their internal lumina and coated with an ultrathin layer of gold and then observed at an acceleration tension of 1 kV in a scanning electron microscope (Hitachi S-4800, Westford, MA, USA).

To observe the samples after SC cultures, samples for SEM were washed in PB 0.1M and fixed in 3.5% glutaraldehyde (Electron Microscopy Sciences, Madrid, Spain) solution for 1 h at 37 °C, post-fixed with 2% OsO_4_ (Electron Microscopy Sciences, Madrid, Spain), and dehydrated. Later, samples were processed in a critical point dryer (critical point values: 328 C, 1100 psi). Samples with conduits were cut longitudinally to expose their internal lumina and observed in a scanning electron microscope. Quantification of the transversal dimension of the structure formed by the PLA fibers and the SC (‘neural cord’) was performed employing SEM images and measured using Image J software.

### 2.6. MTS Assay

To evaluate the cell proliferation inside conduits, MTS assays (CellTiter 96 Aqueous One Solution Cell Proliferation Assay, Promega, Madrid, Spain) were carried out on the unimodular and multimodular conduits (*n* = 5 each). When incorporated into the cells, the MTS was reduced by metabolically active cells at 5 days of SC cell culture in a rate proportional to the number of live cells. After 1 h of incubation with the MTS reagent, the medium was removed, and its absorbance was measured with a Victor Multilabel Counter 1420 spectrophotometer (Perkin-Elmer, Madrid, Spain) at 490 nm.

### 2.7. Immunocytochemistry

After the SC culture or co-culture with DRG explant, SC and neural population were identified, analyzing the expression of different markers by immunofluorescence with fluorescence or confocal laser scanning microscopy: F-actin (ActinRed™ 555 ReadyProbes™ Reagent; for SC) and neuron-specific class III beta-tubulin (Tuj1; for neurons). For confocal microscopy, the conduits were rinsed thoroughly with PB 0.1M and fixed in 4% paraformaldehyde for 20 min. Cells were permeabilized and blocked with 0.1% Triton X-100 (Sigma-Aldrich, Madrid, Spain), 10% FBS in PB 0.1M for 2 h. Conduits were then incubated at 4 °C overnight with primary antibody mouse monoclonal Tuj1 (1/300; Neuromics, Minneapolis, MN, USA). Secondary antibody goat anti-mouse IgG Alexa Fluor^®^ 488 (1/200; Thermo Fisher Scientific, Madrid, Spain) and ActinRed™ 555 ReadyProbes™ Reagent (2 drops/mL; Thermo Fisher, Madrid, Spain) were used for a further 2 h of incubation at room temperature in the darkness. Afterwards, samples were incubated with DAPI (1/5000; Sigma-Aldrich, Madrid, Spain) for 10 min to stain nuclei. It was necessary to make a longitudinal cut of the conduits before performing the immunocytochemistry assay to obtain a complete view of the lumen and the PLA fibers before using a confocal microscope (LEICA TCS SP5, Leica microsystems, Madrid, Spain) or a fluorescent microscope (Nikon Eclipse 80i, Madrid, Spain). The confocal images were processed with an overlay to make a reconstruction of the total length of the conduits. Neurite length on conduit and fibers were measured in the reconstruction of the confocal fluorescent images using Image J software.

In some samples, the conduits were embedded into low gelling temperature agarose (Sigma-Aldrich, Madrid, Spain) after fixation. Then, samples were immersed in 30% sucrose and included in optimal cutting temperature compound (OCT, Thermo Fisher Scientific, Madrid, Spain) at −80 °C. A total of 8 μm transversal sections of the samples were obtained with a cryostat (CM1520, Leica, Madrid, Spain). Transversal sections were immunostained with the same protocol described above.

### 2.8. Gene Expression Experiments

Whole RNA was isolated by a Quick RNA Miniprep kit (ZYMO Research) and quantified by a Q3000 microvolume spectrophotometer (Quawell, San Jose, CA, USA). RNAs were retrotranscribed on a Maxima First Strand cDNA Synthesis Kit with Thermolabile dsDNase (Thermo Fisher Scientific, Madrid, Spain). Real-time quantitative polymerase chain reaction (qPCR) was carried out using the PowerUp SYBR Master Mix (Thermo fisher Scientific, Madrid, Spain) and StepOnePlus™ Real-Time PCR System (Thermo Fisher Scientific, Madrid, Spain). Gene expression of GAP43 was quantified by the ΔΔCt method. Sample values were normalized to the threshold value of the housekeeping gene GAPDH.

### 2.9. Statistical Analysis

Each experiment was performed at least four times unless otherwise noted. For all the experiments, three (SEM and immunocytochemistry assays) or five (MTS assay) independent replicates (*n* = 3 or *n* = 5) of each studied group were employed. Data were expressed as mean ± standard deviation. The Shapiro–Wilk test was used to confirm the data normality on GraphPad Prism 8. Results were analysed by t-student test on normal data and Mann–Whitney test in the opposite case. A 95% confidence level was considered significant. An asterisk * indicates statistically significant differences, indicating a *p*-value below 0.05.

## 3. Results

### 3.1. Manufacture of Uni- and Multimodular Conduits

Unimodular and multimodular HA conduits were obtained with the same procedure as previously described [31,32,34] with 1 mm width, an internal cylindrical channel of 400 μm and variable in length. In Figure 3, we show an unimodular conduit of 12 mm length and a multimodular conduit formed by three individual modules of 4 mm length each, in the fully hydrated stage. A macroscopic view of both conduits in aqueous solution is shown in Figure 3D,E. SEM images of a transversal cut (Figure 3A) of a hyaluronic acid conduit, and longitudinal cuts of the unimodular (Figure 3B) and multimodular (Figure 3C) conduits revealed the lumen of the conduits containing the PLA fibers, laid out parallel to the conduit’s axis. In the multimodular concept, the PLA fibers are the common element to all the individual modules forming the multimodular conduit. Although our present study was restricted to three consecutive modules, the manufacture process allows any number of such modules to be assembled. Thus, tubular scaffolds of any length could be in principle fabricated (Figure 3F). Since the possibility of errors, defects and failures in scaffold manufacture is proportional to their length, the multimodular solution minimizes, for a given length, the possibility of such failures. Furthermore, thanks to the small gaps between individual modules, the multimodular conduit is more apt to curve than an equivalent unimodular conduit (Figure 3E).

### 3.2. Schwann Cells Have Lower Density and Are Less Evenly Distributed in Long-Unimodular Conduits Than in Multimodular Conduits

We seeded rat SC in the lumen of the 18 mm length unimodular conduit and multimodular conduits of three 6 mm length modules and cultured them for 5 days (Figure 4). MTS assay performed after 5 days of SC culture (Figure 4A) shows that the number of cells in unimodular conduits was significantly lower than in the multimodular conduits. A macroscopic view of both conduits after the MTS assay is shown in Figure 3B. The brown color is due to the reduction of MTS reagent, and we could observe areas with a discontinuity in SC density, indicated with a black dash lines box, explaining the higher cell density in the multimodular conduits.

### 3.3. Schwann Cells Grow without Loss of Continuity across the Whole Length of the Multimodular Conduit

Unimodular and multimodular HA conduits were seeded with SC and cultured for 5 days as previously shown. Longitudinal cuts of these samples were analyzed with confocal microscopy (Figure 5). Fluorescent reconstructions in Figure 5A,G, completely spanning the whole length longitudinally cut of unimodular and multimodular conduits, show nuclear staining and F-actin of SC. As previously described in an HA conduit of 6 mm length [31,32,34], the SC form a cylindrical sheath-like tapestry continuously spanning to the whole length of the internal lumen and SC attached to and grew on the PLA fibers too, and completely covered them. We could observe that SC growing in the lumen of the unimodular and multimodular conduits with a total length of 18 mm. SC grew on the PLA fibers at the beginning at 6 mm, 12 mm, and 18 mm in the unimodular sample (Figure 5B–E). Moreover, SC grew at the beginning and the end of multimodular samples and also at the inter-module zone between modules 1 and 2 and the inter-module zone between modules 2 and 3 (Figure 5H–K).

### 3.4. Axons Extend Continuously from End to End of He Multimodular Conduit, Bridging the Inter-Module Gaps

The postnatal P3–4 rat DRG explants were cultured for 21 days at one end of the unimodular and multimodular HA conduit with pre-seeded SC cultured for 5 days (Figure 6) in order to study the axonal extension from the DRG explant, by neuronal staining with Tuj 1 (green). The DRG projecting axons fully invade the lumen of the multimodular conduit (Figure 6), reaching the opposite end of the multimodular 18 mm length conduit, while in the unimodular conduit the maximum length was approximately 11 mm (Figure 6). In the multimodular conduit, fibers pass through the individual modules, axons extended all through the fibers’ length, even without the HA conduit in the inter-module zone between modules 1 and 2 (indicated with m1, m2 in Figure 6) and the inter-module zone between modules 2 and 3 (indicated m2, m3 in Figure 6). The number of axons that reached the end of the multimodular conduit was less than at the beginning, according to the staining intensity (Figure 6).

### 3.5. A ‘Neural Cord’ Grown within and Later Extracted from the Conduit Is Viable for Additional Days in Culture as an Independent Biohybrid Live Structure

Multimodular HA conduits were seeded with SC and cultured for 5 days. Then, we eliminate the HA conduit making a longitudinal cut and keep this structure in the culture for 5 more days. We obtained a ‘neural cord’ formed by the PLA fibers and the associated SC (Figure 7). This ‘neural cord’ is an engineered live biohybrid. Figure 7A shows a scheme of longitudinal and transversal view of the ‘neural cord’ obtained. Figure 7B,B’ show the ‘neural cord’ in culture before fixing. Figure 7C shows a fluorescence complete length reconstruction of the ‘neural cord’, made from a series of fluorescent images of longitudinal sections of a ‘neural cord’ showing SC F-actin staining, and its detail. After the 5 additional days, we could observe that the ‘neural cord’ was intact after manipulation 5 days before to eliminate the HA conduit and also before fixing it. We could prove the stability against manipulation of the ‘neural cord’ without altering it with the SEM reconstruction in Figure 7D, and it is detailed in Figure 7D’, where we could see a compact ‘neural cord’.

Figure 8A,B show SC associated with PLA fibers in multimodular conduit after a culture of 5 and 10 days, respectively. The multimodular conduits were cut longitudinally to obtain an internal view (Figure 8A,B). Figure 8C,D show the ‘neural cord’ after 5 additional days in the culture after eliminating the HA conduit. In Figure 8A,B, we could observe a double structure formed by SC, those SC adhered on the PLA fibers, covering them, and a ‘SC sheath’ templated by the inner surface of the HA conduit lumen that detached from the internal lumen and started to attach to the SC-PLA fibers structure. In Figure 8C,D we could not distinguish these two different cellular structures. The ‘neural cord’ shows a more abundant and compact SC structure in comparison with multimodular conduits after 5 and 10 days of culture. The transversal dimension of the ‘neural cord’ (Figure 8C,D) ranges between 237 µm in the thinnest areas and 285 µm in the thickest areas in comparison with 202 µm and 250 µm for the structure after 5 days of culture (Figure 8A) and 178 µm and 273 µm after 10 days of culture (Figure 8B).

To analyze whether the SC colonized the interior of the ‘neural cord’ in the 5 additional days of culture, we transversally cut the ‘neural cord’. We could observe the internal distribution SC between PLA fibers (Figure 9A,B).

### 3.6. Axons Extend through the ‘Neural Cord’ after Co-Culture with Dorsal Root Ganglion Explants

Preliminary results of the culture of a DRG explant positioned at one end of a ‘neural cord’ seem to confirm that it is indeed possible to grow and guide the axons through it (Figure 10). We have been able to observe the axonal extension with DRG explants for a total of 14 days (10 days with the HA conduits and 4 days without the HA conduits). We maintained the HA conduits for 10 days to ensure the adhesion and growth inside the lumen of the conduits of DRG explants. We performed a short-term culture as a proof of concept to study the axonal extension. The distance covered by the axons was approximately 4 mm long.

## 4. Discussion

Biohybrid approaches for neural tissue engineering seem to have better results in terms of functional recovery in comparison with purely material-based constructs [40]. SC have demonstrated to enhance axonal growth in vitro [41] and to regenerate nerves in combination with implants [35,42], and even in the CNS they proved to support the recovery of spinal cord function [43]. The regeneration of tract-like axon structures demands bridging sometimes long distances directing axon growth in an ordered way.

In previous studies [31,32,34,35], we verified that SC seeded and cultured within HA conduits formed a ‘SC sheath’, a cell cylinder templated by the inner surface of the conduit’s lumen. These SC also grow on PLA fibers when these are placed inside the conduit’s lumen. The combined use of parallel aligned PLA fibers and SC recapitulates directional characteristics of the in vivo physiological situation produced during neural regeneration. We have also verified that this biohybrid construct was capable of sustaining axonal growth from neurons of a DRG explant positioned at one end of the conduit [34]. These axons grew unidirectionally and reached the opposite end of the 6 mm conduits. Real clinical situations face lesions that are usually longer, and the gap to be regenerated is often critical. Frequently, it is not possible to regenerate lesions with gaps > 10–12 mm [9], although it is increasingly common to find promising results in longer lesions, even at critical distances of 14–15 mm [14,44]. Lesions produced in the CNS, such as SCI or TBI, also lead to the loss of guidance through large distances [2]. The need to achieve axon regeneration through larger distances motivates the concept explored here, which focuses on our previous experience and uses it as the ‘unit’ of a multimodular conduit in order to span longer lengths. Several individual conduits (modules) consecutively placed after the other share a common bundle of PLA fibers in their lumen (Figure 3). By changing the number of modules, the conduit can be produced in different lengths (Figure 3E) in a versatile and simple way.

A multimodular concept presents several advantages compared to a single conduit of the same length. Defects during the manufacturing process become more probable the longer the conduit; thus, shorter conduits are manufactured more effectively and the desired final length has no bearing on the manufacturing process. Another advantage of a multimodular concept arises when cells are seeded. We verified that there was a higher cell density in the multimodular conduit than in the equivalent unimodular conduit (Figure 4A), where, moreover, the SC growth was not uniform throughout the lumen of the unimodular conduit (Figure 4B). This fact can be related to the way the SC are seeded in the conduits. In the case of a unimodular conduit, the pipette has only two accessible seeding points, the two ends, and the SC cannot self-organize to completely cover the entire interior of the conduit during the 5 days of culture. This may also have to do with the access of nutrients to the cells, which will also be different in a uni- and a multimodular conduit. In the multimodular conduit each individual module is cell-seeded, and the SC coat the entire lumen of each 6 mm of the individual modules. The non-tight intermodule spacings of the multimodular conduit allow better diffusion of nutrients and results in a uniform coating of SC (Figure 5I,J). In axonal regeneration it has been reported that increased permeability improves the process [45], even being an option to eliminate the conduit cover with promising results [46].

We have shown that the multimodular concept is able to induce the axonal growth from end to end of the 18 mm multimodular conduit (Figure 6B). Human axon growth during peripheral nerve regeneration has a rate of approximately 1 mm/day [47], and this rate is significantly lower in the CNS [48]. DRG axons grow in vitro with rates between 130 and 300 µm/day [49]. In our present study, we can estimate the axonal growth rate in vitro for the multimodular conduit at approximately 600–700 µm/day, a value obtained by dividing the maximum length reached at day 21 by axons through that time. Moreover, the most effective implantable medical devices for functional regeneration currently do not exceed rates of 0.5–1 mm/day [50]. The axons from the DRG seeded at one end of the multimodular conduit, after 21 days, extended across the three individual modules, crossing the two inter-module separation zones devoid of the protection of the HA conduit. Axons were able to reach the farthest end of the last individual module, although in fewer numbers than in the initial area of the construct. The absence of the HA conduit in the inter-module spacings did not prevent SC from growing on the PLA fibers in a continuous and unidirectional manner (Figure 5G–K). In this way, the axons, which grow accompanied by the SC attached to the PLA fibers, were able to grow through the entire length of the multimodular conduit bridging the inter-module gaps. This result, added to the previously mentioned advantages of the modular concept, makes HA modular conduits with PLA fibers and pre-seeded SC an option for the regeneration of long lesions of tract-like structures of the nervous system. Our first partial experience with this kind of implant in vivo has been reported in reference [35]. There the multi-modular concept was scaled to match physiological dimensions of the sciatic nerve in rabbits. After six months, follow-up after surgery regeneration was analyzed. The main conclusions there were (i) the better vascularization of the multi-module implant compared to the unimodular one, and (ii) that the gaps between modules did not represent any obstacle to regeneration, which took place across the whole length of the multi-module implant without loss of continuity. Those in vivo results can be explained as a consequence of the conditions of axon regeneration analyzed in the present study ex vivo.

The hybrid material-cell structure that is formed within the multimodular conduit, composed of the PLA fibers and the SC, becomes an entity and survives when extracted from the HA conduits. This ‘neural cord’ (Figure 7 and Figure 8) has consistency and integrity on its own and can be manipulated without compromising the stability of the structure. The SC in this ‘cord’ survived after excising the HA conduits with a scalpel and being cultured outside the conduit (Figure 7B). A short-term co-culture of the ‘neural cord’ with a DRG explant placed at one of its ends showed that axons sprouting from the neuronal somata in the DRG extended along the ‘cord’ approximately 4 mm. It may be expected that in a longer-term co-culture of SC with DRG explants experiment the ‘neural cord’ will be able to induce longer axonal extensions (Figure 9). In this respect our results are preliminary, and the idea of axon growth into the ‘neural cord’ will probably have to be further optimized before establishing conclusions as to the efficiency of the process and the growth rates.

This ‘neural cord’ represents an ex vivo engineered biohybrid that could act as a template for tract regeneration. It would make it possible to develop directly transplantable structures without conduits when this could be esteemed as an advantage (the SC and the neurotrophic factors they release [24] would be more accessible and thus guide the axonal regrowth across the regenerating site [12,51]); in the case of peripheral nerve regeneration, the ‘neural cord’ could be transplanted employing the microsurgical epineural sheath tube technique [12] or it could be sutured directly to the stumps [52] or introduced in other nerve guidance conduits.

## 5. Conclusions

It has proved possible to induce axonal extensions of at least 18 mm with a multimodular concept that employs biomaterials and cells in complementary functions. It thus becomes possible to consecutively add a number of elementary units to achieve desired regeneration lengths. The main problem to be faced by such a hypothesis, namely the continuity of cell and axon growth across the intermodular gaps, was solved due to a fiber bundle common to the lumina of all modules. Additionally, the fiber + cells complex grown in vitro within the multimodular scaffold can be extracted and handled as an autonomous unit, a ‘neural cord’. It represents an ex vivo engineered hybrid template for induced axon growth that could be transplanted as such or as a part of more complex strategies. Overall, the results open up new perspectives for the problem of regenerating long tract-like axon structures in the nervous system.

## 6. Patents

M. Monleón Pradas, C. Martínez Ramos, L. Rodríguez Doblado, F. Gisbert Roca. Dispositivo modular para regeneración nerviosa y procedimiento de fabricación. ES20210030065 20210127, ES2818424, 2021.

## Figures and Tables

**Figure 1 biomedicines-10-02228-f001:**
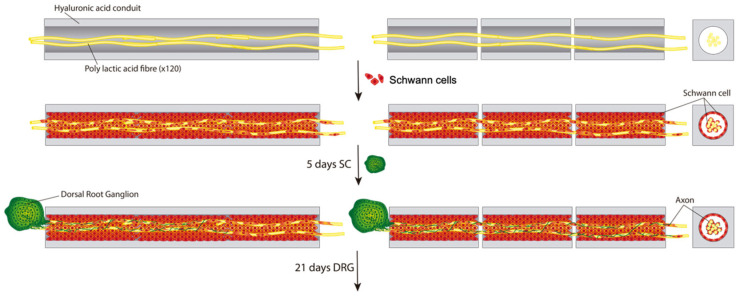
Schematic view of the generation of the long-distance axonal regeneration strategy proposed. Longitudinal and transversal view of a unimodular conduit (**left**) and multimodular conduit (**right**) formed by three individual modules of HA and PLA fibers passing through its lumens. We seeded SC for 5 days and the SC formed a tapestry in the inner lumen and also grew on the PLA fibers. After 5 days of SC culture, we seeded a DRG explant at one end of the unimodular and multimodular conduits and culture for 21 days, expecting the axons of projecting neurons extent through the conduits. We observed the axon growth to prove the effectiveness of promoting directed axon growth in long distances.

**Figure 2 biomedicines-10-02228-f002:**
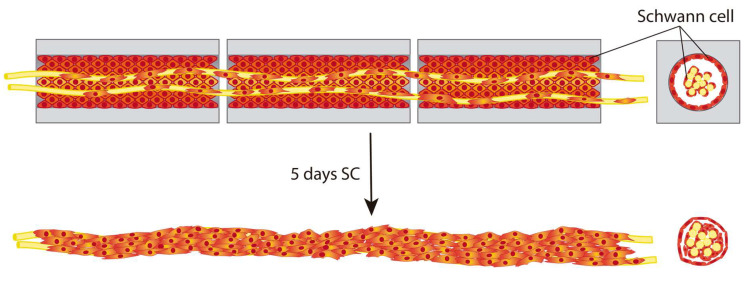
Schematic view of the generation of the ‘neural cord’. Longitudinal and transversal schematic view of ‘neural cord’ after 5 days of SC culture and elimination of the HA conduits, obtaining a live hybrid pseudonerve in vitro for a possible transplantation.

**Figure 3 biomedicines-10-02228-f003:**
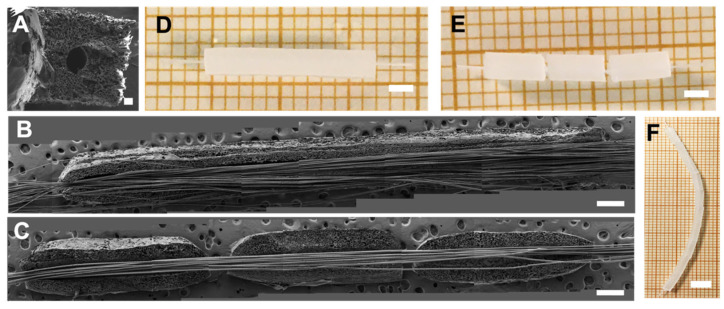
Unimodular and multimodular conduits concept. Scanning electron microscope images from a transversal cut (**A**) of a hyaluronic acid conduit, and a longitudinal cut of (**B**) unimodular and (**C**) multimodular conduits (only the part with de PLA fibers is shown). Macroscopical view of hydrated (**D**) unimodular and (**E**) multimodular conduits, and a (**F**) multimodular conduit formed by 10 modules in order to cover a long and curved distance. The distances between the modules in multimodular conduits is exaggerated for better visualization. Scale bar: 5 mm (**F**), 2 mm (**D**,**E**), 500 μm (**B**,**C**), 100 μm (**A**).

**Figure 4 biomedicines-10-02228-f004:**
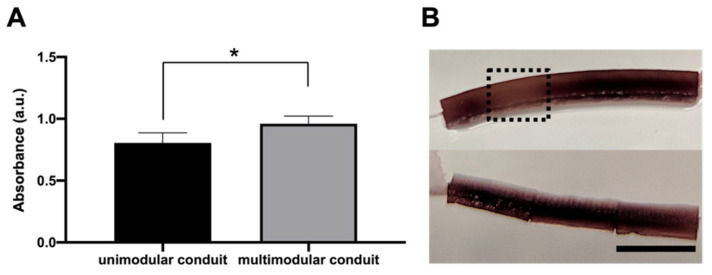
Differences of Schwann cells density between unimodular and multimodular conduits after 5 days. (**A**) MTS assay on SC cultured for 5 days inside unimodular and multimodular conduits. In multimodular conduit cell proliferation was increased. An asterisk * indicates statistically significant differences, indicating a *p*-value below 0.05. (**B**) Macroscopical view of hydrated (**top**) unimodular and (**bottom**) multimodular conduits after MTS assay (reveled with color brown). Zone without cellular continuity in unimodular conduit is indicated with a black dash lines box. Scale bar: 6 mm.

**Figure 5 biomedicines-10-02228-f005:**
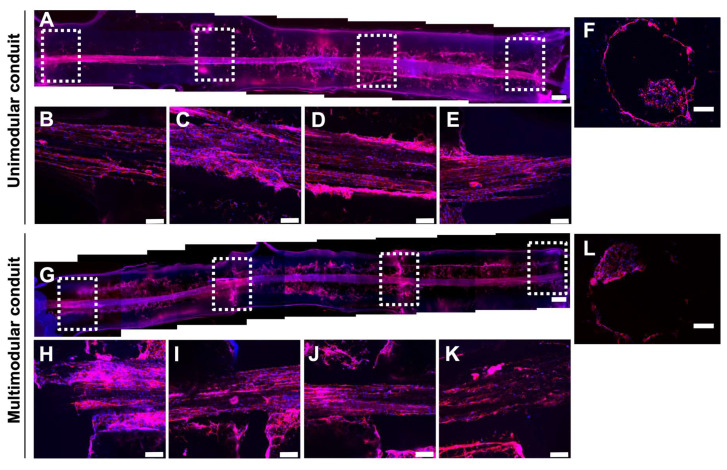
Schwann cells distribution along the whole length and transversal sections of unimodular and multimodular conduits after 5 days. Representative fluorescence reconstruction of the conduit’s complete length of (**A**) unimodular and (**G**) multimodular conduits, after nuclear staining with DAPI (blue) and F-actin of SC staining with ActinRed™ 555 ReadyProbes™ Reagent (red). Confocal fluorescent image of a longitudinal section of unimodular conduit at (**B**) 0 mm, (**C**) 6 mm, (**D**) 12 mm, (**E**) 18 mm, and multimodular conduit at (**H**) 0 mm, (**I**) inter-module zone between modules 1 and 2, (**J**) inter-module zone between modules 2 and 3, (**K**) 18 mm, showing continuity in SC growth even in the absence of the HA conduit. Details in A and G are indicated with white dash lines boxes. Representative fluorescent images of a transversal section at 9 mm long of unimodular conduit (**F**), and multimodular conduit (**L**), after the same staining. Scale bar: 500 μm (**A**,**G**), 100 μm (**B**–**F**,**H**–**L**).

**Figure 6 biomedicines-10-02228-f006:**
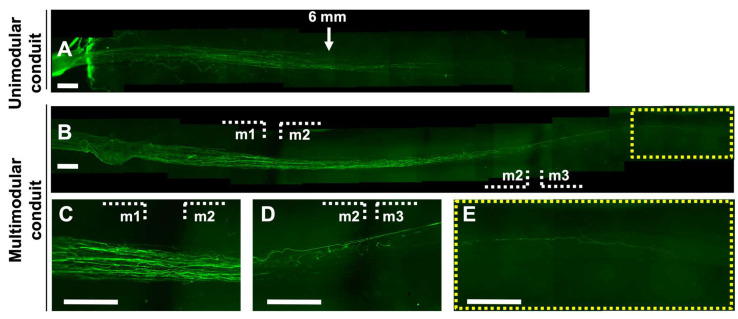
Axon growth in unimodular and multimodular conduits after 21-days dorsal root ganglion explant culture. Representative fluorescence reconstruction of the (**A**) unimodular and (**B**) multimodular conduit’s complete length after neuronal staining with Tuj1 (green), showing greater axonal extension of the multimodular conduit. Fluorescent image of a longitudinal section of (**C**) inter-module zone between modules 1 and 2 (m1, m2), (**D**) inter-module zone between modules 2 and 3 (m2, m3), (**E**) final zone in module 3. White dash lines delimit the modules of the multimodular conduit, and yellow dash lines box delimit the final zone of the module 3 (16–17 mm long from the DRG explant). Scale bar: 500 μm.

**Figure 7 biomedicines-10-02228-f007:**
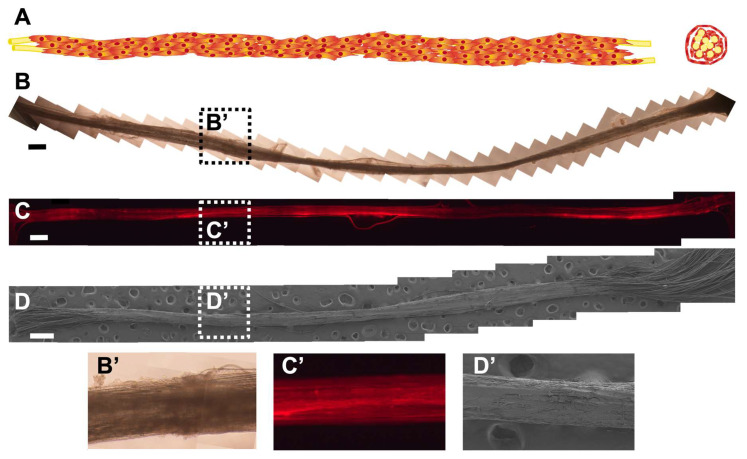
Schwann cells distribution in the ‘neural cord’ after 5 additional culture days. (**A**) Longitudinal and transversal schematic view of ‘neural cord’. (**B**) Representative phase-contrast reconstruction of the ‘neural cord’s’ complete length in culture before fixing. (**C**) Representative fluorescence reconstruction of the ‘neural cord’s’ complete length after F-actin of SC staining with ActinRed™ 555 ReadyProbes™ Reagent (red). (**D**) Representative SEM reconstruction of the ‘neural cord’s’ complete length, showing a compact structure after the manipulation of the sample. Details in reconstructions are indicated with dash lines boxes in (**B’**), (**C’**), and (**D’**). Scale bar: 500 μm.

**Figure 8 biomedicines-10-02228-f008:**
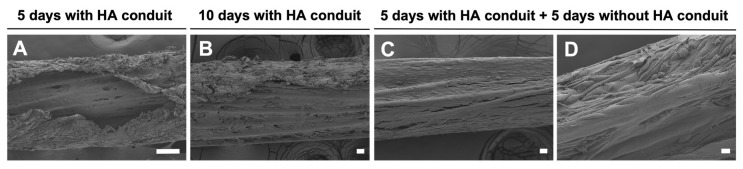
Schwann cells distribution in multimodular conduits and ‘neural cord’. Scanning electron microscopic image from a longitudinal cut in multimodular conduits after (**A**) 5-days culture and (**B**) 10-days culture. (**C**,**D**) Scanning electron microscopic image of the ‘neural cord’ after 5 days of culture outside the HA conduit. Scale bar: 100 μm (**A**), 20 μm (**B**,**C**), 10 μm (**D**).

**Figure 9 biomedicines-10-02228-f009:**
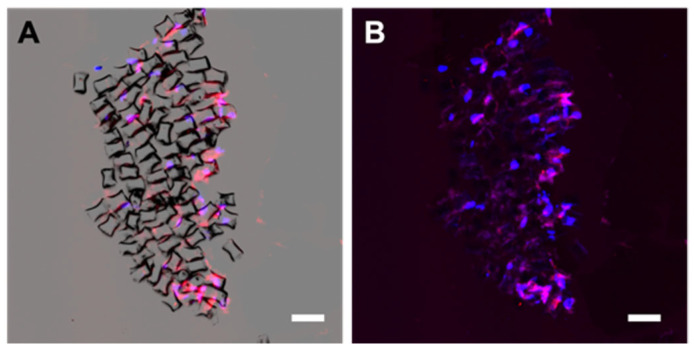
Schwann cells distribution in a transversal cut of the ‘neural cord’. Representative fluorescent images of a transversal section of the ‘neural cord’ after nuclear staining with DAPI (blue) and F-actin of SC staining with ActinRed™ 555 ReadyProbes™ Reagent (red) with (**A**) and without (**B**) the bright field view. Scale bar: 100 μm (**A**,**B**).

**Figure 10 biomedicines-10-02228-f010:**
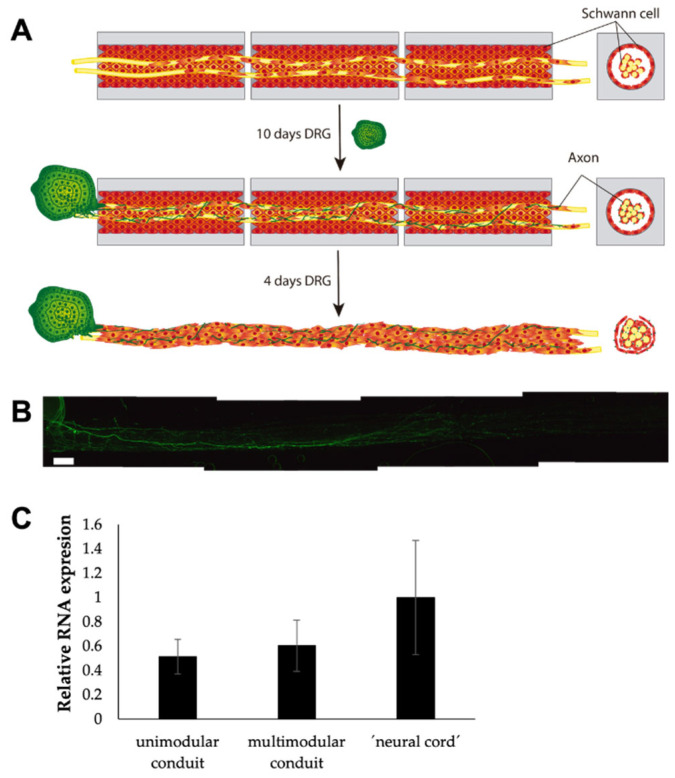
Axon growth in ‘neural cord’ after o-culture with dorsal root ganglion explant. (**A**) Longitudinal and transversal schematic view of obtention of the ‘neural cord’ and the co-culture with a dorsal root ganglion explant. (**B**) Representative confocal reconstruction of the first 4–5 mm of longitudinal sections of the ‘neural cord’ showing neuronal staining with Tuj 1 (green) after 10 days with HA conduits and 16 additional days without the HA conduits. (**C**) Analysis of relative mRNA expression of GAP43. Scale bar: 500 μm.

## Data Availability

All the data generated in this research are included in the manuscript.

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
