# Peer review of "Multimodular Bio-Inspired Organized Structures Guiding Long-Distance Axonal Regeneration"

_biomedicines, 2022, doi:10.3390/biomedicines10092228_

Round 1

Reviewer 1 Report

In the authors previous studies, they showed that the hyaluronic acid conduits with PLA microfibers could support and promote Schwann cell growth for axon regeneration. However, typical biomaterial conduits are not long enough to conduct axons regeneration. Thus, this paper aimed to study the modulation of the long conduits on SCs growing and found that the multimodular conduit could induce a better axonal extensions of at least 18mm, which is pretty promising as an ex vivo engineered biohybrid for axon regeneration. It’s a well-written paper with comprehensive introduction of the topic. Much of the results were well described and the discussion is convincing. However, there are still some confusing points which need further explanation or discussion. Please see the comments below:

1.     It’s a little bit complicated to understand the ex-vivo systems for the SCs cultured in conduits, illustrations or photos should help for better understanding and improve the reproducibility.

2.     The schematic view of Figure 1 and Figure 2 has some overlaps. I recommend re-arrange those figures or combine them.

3.     It’s very interesting that the unimodular conduit did not work well compared with multimodular conduit. Is that just because you put cells in each end of the modules but only two ends of unimodular conduit?

4.     I did not see difference on cell staining/distribution between unimodular or multimodular conduit by Figure 5. The legend of Figure 5 has a mistake (Line 324, “Details in A and F” should be “A and G”). Also, please indicate the position of the transversal section (F and L) in A or G.

5.     How did you determine the number of axons in the conduits as you according to Figure 6?

6.     Improve the quality of Figure 10C with Y-axis title and significant analysis. What’s the group means for 1*18, 3*6 and 4-7-7?

7.     The references are too old, please cite more recently published researches and discuss more about the current trends in this field.  

Author Response

Valencia, 1st September 2022

Dear reviewer,

Please find enclosed the new electronic version of the paper entitled Multimodular bio-inspired organized structures guiding long-distance axonal regeneration” by Laura Rodriguez Doblado, Cristina Martínez Ramos, and Manuel Monleón Pradas submitted to be considered for publication in Biomedicines.

Below we respond to the comments of the reviewers made after the first submission of the manuscript.

All modifications made to the manuscript based on the reviewers' suggestions are underlined in yellow in the revised manuscript.

REVIEWER 1

In the authors previous studies, they showed that the hyaluronic acid conduits with PLA microfibers could support and promote Schwann cell growth for axon regeneration. However, typical biomaterial conduits are not long enough to conduct axons regeneration. Thus, this paper aimed to study the modulation of the long conduits on SCs growing and found that the multimodular conduit could induce a better axonal extensions of at least 18mm, which is pretty promising as an ex vivo engineered biohybrid for axon regeneration. It’s a well-written paper with comprehensive introduction of the topic. Much of the results were well described and the discussion is convincing. However, there are still some confusing points which need further explanation or discussion. Please see the comments below:

  1. It’s a little bit complicated to understand the ex-vivo systems for the SCs cultured in conduits, illustrations or photos should help for better understanding and improve the reproducibility.

It has been changed as suggested.

  1. The schematic view of Figure 1 and Figure 2 has some overlaps. I recommend re-arrange those figures or combine them.

Figures 1 and 2 highlight the two different approaches discussed in this article, which is why we have thought it convenient to separate them. Figure 1 schematizes the axonal growth process in the multimodular duct, while Figure 2 illustrates the generation of the ‘neural cord’, only with the SC, as a possible transplantable structure.

  1. It’s very interesting that the unimodular conduit did not work well compared with multimodular conduit. Is that just because you put cells in each end of the modules but only two ends of unimodular conduit?

The way we seed cells inside the unimodular and multimodular conduits affects the success of the different conduits in maintaining the cellular structure inside, but is not the only reason for the best performance of the multimodular conduit. The intermodule spacings of the multimodular conduit allow better diffusion of nutrients and result in a uniform coating of SC.

  1. I did not see difference on cell staining/distribution between unimodular or multimodular conduit by Figure 5. The legend of Figure 5 has a mistake (Line 324, “Details in A and F” should be “A and G”). Also, please indicate the position of the transversal section (F and L) in A or G.

Figure A shows no SCs in the first 6 mmof the lumen of the HA conduit, with SCs appearing practically only on the PLA fibers.

  1. How did you determine the number of axons in the conduits as you according to Figure 6?

The number of axons was determined according to the staining intensity, as is now described in the manuscript.

  1. Improve the quality of Figure 10C with Y-axis title and significant analysis. What’s the group means for 1*18, 3*6 and 4-7-7?

It has been changed as suggested.

  1. The references are too old, please cite more recently published researches and discuss more about the current trends in this field.  

New references have been added and old ones removed. In addition, more has been discussed about trends in the field of nerve repair.

Yours sincerely,

Manuel Monleón Pradas, Corresponding Author

Centre for Biomaterials and Tissue Engineering. Universidad Politécnica de Valencia 46022, Spain

Reviewer 2 Report

Dear editor,

The manuscript (biomedicines-1888259) reports development of a modular nerve conduit composed of HA incorporating PLA fibers and Schwann cells. The study is interesting and the system has been thoroughly characterized. However, I found a similar publication by the authoring team in Biomedicines (https://doi.org/10.3390/biomedicines10050963). Trivially, the first question coming to mind is that what are the differences between these two studies? These differences should be discussed in Introduction. My other major comments include:

1- To construct the conduit, 3D bioprinting could be also applied. What would be the advantages of the process the authors followed compared to 3D bioprinting of the cell laden hydrogel conduit?

2- What are the advantages of HA compared to other hydrogel materials such as alginate, gelatine, etc.?

3- SEM, 10 kV could be a high accelerating voltage. The conduit material was not degraded?

4- Figure 3A and B are poor quality and do not show the tubular being of the conduits. Moreover, a scale bar could help the reader better understand the dimensions of the conduit.

5- Figure 3C&D; there are plenty of pores in the conduit wall. Any impact from these defects on the overall performance of the conduit? How their number could be minimized during the fabrication process?

6- How much is the elastic modulus of the conduits? Does it suit the target application?

7- Figure 4A, Y-axis unit is a.u., and 3.2 headline, venly should be corrected as evenly.

8- What is the fate of PLA fibers? PLA is well-known to release acidic by-products. This acidification can not adversely affect the cell behaviour?

9- How different would be the performance of the conduits (uni or multimodular) in vitro and in vivo?

Author Response

Valencia, 1st September 2022

Dear reviewer,

Please find enclosed the new electronic version of the paper entitled Multimodular bio-inspired organized structures guiding long-distance axonal regeneration” by Laura Rodriguez Doblado, Cristina Martínez Ramos, and Manuel Monleón Pradas submitted to be considered for publication in Biomedicines.

Below we respond to the comments of the reviewers made after the first submission of the manuscript.

All modifications made to the manuscript based on the reviewers' suggestions are underlined in yellow in the revised manuscript.

REVIEWER 2

The manuscript (biomedicines-1888259) reports development of a modular nerve conduit composed of HA incorporating PLA fibers and Schwann cells. The study is interesting and the system has been thoroughly characterized. However, I found a similar publication by the authoring team in Biomedicines (https://doi.org/10.3390/biomedicines10050963). Trivially, the first question coming to mind is that what are the differences between these two studies? These differences should be discussed in Introduction. My other major comments include:

The study mentioned (https://doi.org/10.3390/biomedicines10050963) is complementary to the present one. In it, we rescaled the present concept to adapt it to a lesion model of sciatic nerve in rats, and followed nerve regenration in vivo. A new paragraph referring to this study has been included in the introduction of the present manuscript.

1- To construct the conduit, 3D bioprinting could be also applied. What would be the advantages of the process the authors followed compared to 3D bioprinting of the cell laden hydrogel conduit?

Bioprinting with cells as a component of the process would probably deliver a completely different end result. The development of a Schwann-cell sheet that coats the inner surface of the conduit’s lumen is a process that encompasses different stages (cell adhesion, proliferation, cell-cell contact formation, spatial reorganization), all of them in the presence of a fixed material surface onto with it takes place. A simultaneous formation through printing of this material-cell layer structure is unlikely to take place. A 3D-printer without cells can be employed to produce conduits, but then the gradient density of the inner surface (obtained in our case through contact with the hydrophobic mould surface) would have to be sacrificed (for an analysis of the wall structure of the conduits, see the reference Vilariño-Feltrer, G.; Martínez-Ramos, C.; Monleón-De-La-Fuente, A.; Vallés-Lluch, A.; Moratal, D.; Barcia Albacar, J.A.; Monleón Pradas, M. Schwann-Cell Cylinders Grown inside Hyaluronic-Acid Tubular Scaffolds with Gradient Porosity. Acta Biomater. 2016, 30, 199–211, doi:10.1016/j.actbio.2015.10.040.).

2- What are the advantages of HA compared to other hydrogel materials such as alginate, gelatine, etc.?

We believe that other hydrogels might perform equally well, but we haven’t checked this. An important point with other kinds of hydrogels would be to ascertain whether the same type of cell coating with Schwann cells obtained.

3- SEM, 10 kV could be a high accelerating voltage. The conduit material was not degraded?

The images were taken using a voltage of 1kV. This has been changed in the manuscript.

4- Figure 3A and B are poor quality and do not show the tubular being of the conduits. Moreover, a scale bar could help the reader better understand the dimensions of the conduit.

It has been changed as suggested to show the tubular being of the conduit. However, the scaffold is defined in detail in previous articles cited in this manuscript:

- Vilariño-Feltrer, G.; Martínez-Ramos, C.; Monleón-De-La-Fuente, A.; Vallés-Lluch, A.; Moratal, D.; Barcia Albacar, J.A.; Monleón Pradas, M. Schwann-Cell Cylinders Grown inside Hyaluronic-Acid Tubular Scaffolds with Gradient Porosity. Acta Biomater. 2016, 30, 199–211, doi:10.1016/j.actbio.2015.10.040.

- Doblado, L.R.; Martínez-Ramos, C.; García-Verdugo, J.M.; Moreno-Manzano, V.; Pradas, M.M. Engineered Axon Tracts within Tubular Biohybrid Scaffolds. J. Neural Eng. 2021, 18, 0460c5, doi:10.1088/1741-2552/ac17d8.

Unfortunately, we could not get images of better quality because of the nature of the scaffolds. We decided to keep them because they provide valuable information on the macrostructure of the conduits. For a better understanding of the images, we have added the scale bars

5- Figure 3C&D; there are plenty of pores in the conduit wall. Any impact from these defects on the overall performance of the conduit? How their number could be minimized during the fabrication process?

We do not regard the porous structure of the conduit’s matrix as a defect; on the contrary, they are functional to their performance. This aspect of the conduit’s structure was discussed in our referenceVilariño-Feltrer, G.; Martínez-Ramos, C.; Monleón-De-La-Fuente, A.; Vallés-Lluch, A.; Moratal, D.; Barcia Albacar, J.A.; Monleón Pradas, M. Schwann-Cell Cylinders Grown inside Hyaluronic-Acid Tubular Scaffolds with Gradient Porosity. Acta Biomater. 2016, 30, 199–211, doi:10.1016/j.actbio.2015.10.040.

The porosity of the conduit’s inner wall of the lumen permits the exchange of oxygen and desired molecules but not the migration of the cells across the scaffold structure.

6- How much is the elastic modulus of the conduits? Does it suit the target application?

The elastic modulus of these conduits are around 3 kPa (Gisbert Roca F, Lozano Picazo P, Pérez-Rigueiro J, Guinea Tortuero GV, Monleón Pradas M, Martínez-Ramos C. Conduits based on the combination of hyaluronic acid and silk fibroin: Characterization, in vitro studies and in vivo biocompatibility. Int J Biol Macromol. 2020 Apr 1;148:378-390. doi: 10.1016/j.ijbiomac.2020.01.149). The applications do not impose any stringent requirement on this parameter since the conduits do not have to withstand significant mechanical stresses.

7- Figure 4A, Y-axis unit is a.u., and 3.2 headline, venly should be corrected as evenly.

It has been changed as suggested.

8- What is the fate of PLA fibers? PLA is well-known to release acidic by-products. This acidification can not adversely affect the cell behaviour?

Like other polyesters, PLA is known to undergo chain scission hydrolytically, and the lactic acid byproducts are bioresorbable. PLA was chosen because of the now already long experience with this material in human implants and sutures.

9- How different would be the performance of the conduits (uni or multimodular) in vitro and in vivo?

In vitro results can be extrapolated to in vivo situations only with great care, given the great difference in complexity of the environment in each case. Our first partial experience with this concept in vivo has been reported in Roca, F.G.; Santos, L.G.; Roig, M.M.; Medina, L.M.; Martínez-Ramos, C.; Pradas, M.M. Novel Tissue-Engineered Multimodular Hyaluronic Acid-Polylactic Acid Conduits for the Regeneration of Sciatic Nerve Defect. Biomedicines 2022, 10, 963. https://doi.org/10.3390/biomedicines10050963, for a sciatic nerve defect model in rabbits. The main conclusions there were (i) the better vascularization of the multi-module implant compared to the unimodular, and (ii) that the gaps between modules didn’t represent any obstacle to regeneration, which took place across the whole length of the multi-module implant. Those in vivo results can be seen as a consequence of the axon regeneration analyzed in the present study in ex vivo conditions. A paragraph mentioning this study has been introduced in the manuscript.

Yours sincerely,

Manuel Monleón Pradas, Corresponding Author

Centre for Biomaterials and Tissue Engineering. Universidad Politécnica de Valencia 46022, Spain

Round 2

Reviewer 1 Report

Can be accepted at present form.

Reviewer 2 Report

The manuscript is publishable now.